# FairSync: Ensuring Amortized Group Exposure in Distributed Recommendation Retrieval

## ABSTRACT

Driven by considerations of fairness, business, and balanced development needs, the recommender system (RS) often necessitates ensuring that certain groups have a minimum level of exposure within a period of time. For example, RS platforms often have the demand to ensure adequate exposure for new providers or specific categories of items according to their needs. Modern industry RS usually adopts a two-stage pipeline: stage-1 (retrieval stage) retrieves hundreds of candidates from millions of items distributed across various servers, and stage-2 (ranking stage) focuses on presenting a small-size but accurate selection from items chosen in stage-1. Existing efforts for ensuring amortized group exposures focus on stage-2, however, stage-1 is also critical for the task. Without a high-quality set of candidates, the stage-2 ranker cannot ensure the required exposure for the selected groups. Previous fairness-aware works designed for stage-2 typically require accessing and traversing of all items. In stage-1, however, millions of items are distributively stored in servers, making it infeasible to traverse all of them. How to ensure the global amortized group exposures in the distributed retrieval process is a challenging question. To address this issue, we introduce a model named FairSync, which transforms the problem into a constrained distributed optimization problem. Specifically, FairSync resolves the issue by moving it to the dual space, where a central node aggregates historical fairness data into a vector and distributes it to all servers. In theory, by utilizing both local and distributed searching techniques, we can ensure the necessary global amortized exposures. To trade-off the efficiency and retrieval accuracy, the gradient descent technique is used to periodically update the parameter of the dual vector. The experiment results on two public recommender retrieval datasets showcased that FairSync outperformed all the baselines, achieving the desired minimum level of exposures while maintaining a high level of retrieval accuracy.

## ACM Reference Format:

Anonymous Author(s). 2023. FairSync: Ensuring Amortized Group Exposure in Distributed Recommendation Retrieval. In *Proceedings of ACM Conference (Conference'17)*. ACM, New York, NY, USA, 10 pages. https://doi.org/10.1145/nnnnnnn.nnnnnnn

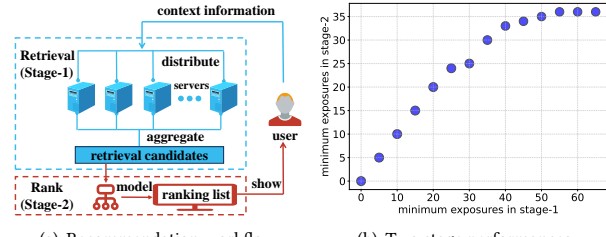

(a) Recommendation workflow     (b) Two stage performances

**Figure 1: (a) The two stage pipelines of recommender system, including retrieval (stage-1) and ranking (stage-2). (b) Simulations depicting the changes for the minimum exposures across two stages.**

## 1 INTRODUCTION

In recent times, the need for fair recommender systems (RS) has gained prominence in industrial requirements [22, 23, 35]. Among these requirements, RS platform has the demands of guaranteeing that specific groups achieve a minimum level of exposure to items within a defined time period, aligning with the perspective of amortized group max-min fairness (MMF) [3, 19, 25, 28, 40, 43]. For instance, certain studies propose to ensure minimum item exposures for new providers [5, 31, 40, 41] for attracting providers to join, while others focus on enhancing the visibility of specific item categories [47, 48] for promoting certain festivals. Such "minimum wage policy" [34] significantly contributes to the enhancement of RS, fostering the creation of a more equitable and robust ecosystem.

In modern RS, two-stage pipelines have been widely adopted, as shown in Figure 1 (a). The primary objective of stage-1 (retrieval) is to efficiently generate a small set of candidates from millions of items in a distributed manner within milliseconds [6, 9, 21, 26] while stage-2 (ranking) more accurately deals with the candidates selected in stage-1 and generates the final recommendations (usually single-digit items) [40, 48]. Regarding ensuring minimum group exposures in RS, most existing studies [5, 10, 29, 31, 39, 40, 48] primarily concentrated on stage-2.

Existing studies revealed that the fairness task in stage-2 can be compromised if stage-1 failed to retrieve a sufficient number of items [38]. We also conduct a simulation to examine how the minimum exposure in stage-1 affects the minimum exposure in stage-2. Specifically, we conduct a simulation using Amazon [1] dataset to assess the minimum exposure of groups across two stages. In stage-1, we leverage the YoutubeDNN [9] model and employ a rule-based method to regulate the retrieved exposures of item categories. In stage-2, we implement an oracle ranking model, ensuring the attainment of the highest minimum exposure of groups. The x-axis represents the minimum exposures of item categories in stage-1,

[1]http://jmcauley.ucsd.edu/data/amazon/

while the y-axis corresponds to the exposures of these categories in stage-2. The results reported in Figure 1 (b) indicate that there exists a robust positive correlation between the minimum group exposure of the two stages. In the simpler terms, if stage-1 is unable to retrieve required item categories effectively, it will also pose challenges for stage-2 in ensuring the exposure of certain groups.

Though critically important, existing approaches designed for stage-2 [10, 29, 31, 39, 40] cannot be directly applied to stage-1 because they ensure the amortized group exposures by traversing all items and adjusting exposures through aggregating information. During stage-1, however, traversing all items (usually millions of items) is infeasible because (1) these items are distributively stored at different servers, and (2) accessing millions of items causes substantial computational cost. While some heuristic approaches have employed strict rules or adjusted group weights, they still lack the capacity to effectively address the challenge.

In this paper, we introduce a novel model named FairSync, which can ensure the minimum amortized group exposure requirements in the retrieval stage of RS. FairSync converts the problem into a constrained distributed optimization and addresses the problem by transferring it to the dual space. In such space, we aggregate past fairness information into a vector and distribute it to servers. Based on the vector, each server independently conducts the item retrieval. Theoretical analysis shows that, even with local and distributive search, FairSync can still achieve global fairness.

In particular, the RS platform first sets a target to ensure that every group attains a minimum level of exposure. Then we approach the problem by formulating it as a distributed resource allocation problem [2, 40], with the constraint of required exposures. Subsequently, we can transform the constrained optimization problem into an unconstrained dual problem. Then, a constructed dual vector, stored the past fairness information, is combined with user embeddings to form a query vector. Each item, along with its embeddings, was concated with the group embedding to form the new item embeddings which will be distributed across servers. After that, each server conducts KNN search for identifying candidate items within milliseconds by using the dense retrieval [44] architecture. Finally, the outcomes are aggregated into a set of candidate items for stage-2. As for the learning procedure, we employ the gradient descent method [18] to update the parameters of dual vector periodically to trade-off the efficiency and effectiveness.

We summarize the major contributions of this paper as follows:

(1) We emphasize the critical importance of incorporating the assurance of minimum exposure for specific groups into the distributed stage-1 (retrieval) of RS.

(2) We introduce a model named FairSync which is tailored to meet the distributed, efficient, and online demands of the prevalent dense retrieval architecture in stage-1 of RS.

(3) The experimental results on two publicly available large-scale recommendation datasets clearly demonstrate that FairSync outperforms the baseline models, attaining the desired minimum level of exposures while preserving a high level of retrieval accuracy.

## 2 RELATED WORK

Fairness has emerged as a prominent research theme within recommender systems. In this realm, two predominant aspects are often explored: individual fairness [24, 27], which concentrates on equitable treatment for individuals, and group fairness, which categorizes items into various groups such as providers [5, 29, 31, 39–41], and item categories [12, 38]. In group fairness, there are usually two criteria. One is egalitarian proposes [12, 27, 29, 38, 39], which aims to equalize the outcome of different groups, another is Rawl's principle [19], which aims to improve the utility of worst-off groups [5, 31, 40, 41]. In real application of RS, amortized fairness [4, 5, 31, 40, 41] is more realistic, which achieves fairness over a period of time, rather than enforcing it strictly on an a single ranking list. In our research, we mainly focused on the amortized group max-min fairness, which is used to support new providers or enhance the visibility of specific item categories.

In RS, there are many methods proposed to alleviate amortized group MMF. FairRec [31] and its extension FairRec+ [5] proposed a offline recommender model to guarantee equal frequency for all items in a series of ranking lists. Yang and Ai [42] proposed a marginal optimizing approach to conduct amortized MMF in the learning-to-rank process. TFROM [39] and CP-Fair [29] proposed a Linear Programming (LP)-based method to ensure the group fairness, see also [3, 10, 25, 43]. P-MMF [40], LTP-MMF [41] proposed a online mirror gradient descent to improve worst-off provider's exposures in the dual space. Nonetheless, all of these proposals have been introduced within the context of stage-2 scenarios, making them impractical for application in stage-1 due to their substantial computational overhead.

In large RS, the significance of stage-1 (retrieval) cannot be overstated, as the performance of stage-2 is heavily reliant on it [6, 20, 21, 38, 45]. There are also some work proposed inspiring approaches to solve fairness issue in the stage-1. Wang and Joachims [38] proposed a uncertainty quantification approach to control the threshold of each retrieval channels in one retrieval process. Hao et al. [14], Rastegarpanah et al. [32] proposed a fairness-related matrix factorization method to adjust the weight the retrieval model. In resource allocation, Balseiro et al. [2], Cheung et al. [8] proposed a mirror-descent method to solve in the dual space. However, these methods either fall in addressing amortized group max-min fairness well or are unsuitable for implementation within the retrieval systems that require distributive, efficient, and online capabilities.

## 3 PROBLEM FORMULATION

In RS, let $\mathcal{U}, \mathcal{I}$ be the set of users and items and each item $i \in \mathcal{I}$ is associated to a unique group $g \in \mathcal{G}$. The set of items associated with a specific group $g$ is denoted as $\mathcal{I}_g$. When a specific user $u \in \mathcal{U}$ accesses the retrieval system, the system will retrieve items from distributed servers and aggregate them into a list of candidate items with a predefined size of $K$, denoted by $L_K(u) \in \mathcal{I}^K$, which is then prepared for stage-2 for detailed ranking.

In real-world applications, the users arrive at the RS sequentially. Assume that at time $t$, user $u_t$ arrives. The RS aims to task with ensuring that the exposure of a specific group $g$ remains at or exceeds a threshold of $m_g$ throughout the entire time horizon from $t = 1$ to $T$, all the while optimizing to retain the enough relevant items within a single candidate retrieval list. In the same time, we require an online solution, where at time step $t$, the RS responds to a request from user $u_t$ by providing a candidate list without waiting

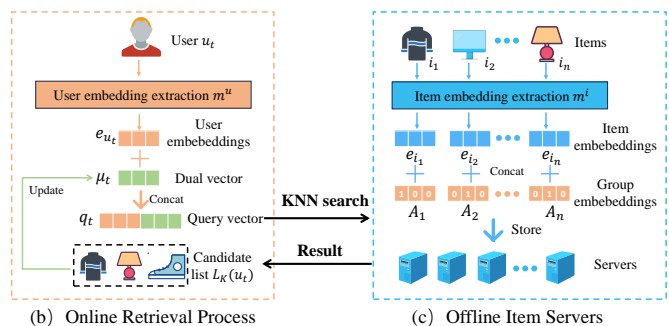

(a) Illustrative Example        (b) Online Retrieval Process        (c) Offline Item Servers

**Figure 2: FairSync Framework. Sub-figure (a) illustrates an example to show the intuitive example of how FairSync works. Sub-figure (b) illustrates that the online retrieval process when user $u_t$ arrives, while sub-figure (c) depicts the offline item embeddings in the dual space construction process.**

for input from a second user $u_{t+1}$. An online retrieval algorithm $h$ produces a real-time decision candidates $L_K(u_t)$ based on the current user $u_t$ and the previous history $\mathcal{H}_{t-1} = \{u_s, L_K(u_s)\}_{s=1}^{t-1}$:

$$L_K(u_t) = h(u_t \mid \mathcal{H}_{t-1}, \mathcal{M}),$$

where $\mathcal{M} = \{m_g | g \in \mathcal{G}\}$ is the factor set by the platform.

State-of-the-art recommender retrieval models [6, 20, 21, 45] usually employ the distributive dense retrieval architecture, wherein an item $i$ is represented as an embedding $e_i \in R^d$ using complex neural networks, such as transformers [37]. These embeddings are indexed on each server $S_n, n \in [1, 2, \cdots, M]$ in a distributed manner, with $d \in N^+$ being the predefined dimension and $M$ is the server number. For the user $u$, a simple network is employed to represent them as an embedding $e_u \in R^d$, typically utilizing their historical browsing information in state-of-the-art systems. The user-item relevance score $r_{u,i}$ is calculated as the distance between $e_i$ and $e_u$ locally in each server $S_n$. The retrieval model's objective is to identify candidate items whose embeddings $e_i$ are in close proximity to the embedding of the user $e_u$, i.e., finding the highest possible relevance scores $r_{u,i}$ in the candidate list for the stage-2 ranking process.

Generally, the RS will establish the offline index [17] for items to efficiently search the desired ones from each server. Previous research [10, 31, 40] focus on ensuring amortized group exposure $\mathcal{M}$ by traversing all items $i$ and their corresponding groups $g$, where $i \in \mathcal{I}_g$. However, in a distributed dense retrieval architecture, these methods are no longer suitable.

## 4 OUR APPROACH

In this section, we will introduce our approach FairSync. To begin with, we will introduce the distributed dense retrieval architecture of stage-1. Then, we will frame the retrieval process as a distributed and constrained resource allocation problem and subsequently formulate its dual problem. After that, we propose to tackle this problem in the dual space integrating it with the distributive dense retrieval architecture and employing the gradient descent [1] technique to efficiently approach a solution.

### 4.1 Distributed Dense Retrieval Architecture

In the mainstream recommender retrieval architectures, the primary objective is to identify items whose embeddings $e_i$, are in close proximity to the embedding of the user $e_u$ distributively. Formally, the problem can write as:

$$L_K(u) = \underset{L \subset \{1,2,\ldots,|\mathcal{I}|\}, |L|=k}{\arg\min} \sum_{i \in S_n, \forall n} d(e_u, e_i), \qquad (1)$$

where $L$ is the set of indices of the $K$ nearest neighbors, $d(e_u, e_i)$ is the distance between embedding $e_u, e_i$, $i$-th the commonly used distance metric being the dot-product locally on each server, i.e.

$$d(e_u, e_i) = -e_u^\top e_i,$$

and the $e_u$ and $e_i$ are calculated by a complex model, such as Deep Neural Network [9], Recurrent Neural Network [16], Capsule Network [21], i.e.

$$e_u = m^u(u), \quad e_i = m^i(i),$$

where $m^u(\cdot)$ and $m^i(\cdot)$ are two embedding extraction networks.

Typically, the item embeddings $e_i$ are pre-calculated and distributively indexed on servers [17], whereas the user embedding $e_u$ requires online inference using complex recommendation models, see [6, 9, 21, 45]. In real application [6], Equation (1) is computed by performing KNN search in the embedding space efficiently.

### 4.2 Dual Space of Retrieval

After the platform gives the minimum exposure requirement for each group, i.e., requiring the exposure of a specific group $g$ to remain at or exceed a threshold of $m_g$ throughout the entire time horizon from $t = 1$ to $T$. Therefore, we write the equation (1) as a distributed resource allocation problem:

$$
\begin{aligned}
\max_{x_{u_t,i}} \quad & \sum_{t=1}^{T} \sum_{i \in S_n, \forall n} x_{u_t,i} r_{u_t,i} \\
\text{s.t.} \quad & \sum_{i \in S_n, \forall n} x_{u_t,i} = K, \quad \forall t \in [1, 2, \ldots, T] \\
& r_{u_t,i} = -d(e_{u_t}, e_i) \\
& e_g = \sum_{t=1}^{T} \sum_{i \in \mathcal{I}_g} x_{u_t,i}, \quad \forall g \in \mathcal{G} \\
& e_g \geq m_g, \quad \forall g \in \mathcal{G} \\
& x_{u_t,i} \in \{0, 1\}, \forall t \in [1, 2, \ldots, T], i \in \mathcal{I}
\end{aligned}
\qquad (2)
$$

where $e_g$ can be seen as the total number of exposed items of group $g$, accumulated over the period 1 to $T$, $x_{u_t,i} \in \{0, 1\}$ is the decision vector for user $u_t$. Specifically, for each item $i$, $x_{u_t,i} = 1$ if it is added to the candidate list $L_K(u_t)$, otherwise $x_{u_t,i} = 0$.

THEOREM 1. *The dual problem objective $W^{Dual}$ of Equation (2) can be write as*

$$\min_{\boldsymbol{\mu}} \left[ \sum_{t=1}^{T} \sum_{k=1}^{K} (r_{u_t,i} - A\boldsymbol{\mu})_{[k]} + \sum_{g \in \mathcal{G}} m_g \boldsymbol{\mu}_g + \max_g \{\boldsymbol{\mu}_g\}(TK - \sum_{g \in \mathcal{G}} m_g) \right],$$

(3)

*where we can have a dual variable $\boldsymbol{\mu} \in \mathbb{R}^{|\mathcal{G}|}$, $A \in \mathbb{R}^{|\mathcal{I}| \times |\mathcal{G}|}$ is the item-group adjacent matrix, and $A_{ig} = 1$ indicates item $i \in \mathcal{I}_g$, and 0 otherwise. Moreover, the dual problem is a strong-dual problem, that is the optimal value of Equation (3) is the same as Equation (2).*

REMARK 1 (DISTRIBUTED SOLUTION IN DUAL SPACE). *After the transformation of the original problem (Equation (1)) into its dual form (Equation (1)), we can convert the problem into an unconstrained optimization problem, simplifying the optimization process significantly. We can also observe that different items are independent with each other. Therefore, the problem can be effectively solved in a distributed manner. The value $\boldsymbol{\mu}$ can be regarded as the accumulated exposure information and be distributed to each server during the retrieval process.*

REMARK 2 (SMALL COMPUTATIONAL COSTS). *The original problem (Equation (1)) is computationally intensive due to its nature as a constraint integral linear problem, and it involves a vast variable space of size $T \times |\mathcal{I}|$ given that the retrieval process may encompass millions of items. However, in the dual problem (2), we observe that the variable size has been significantly reduced to $|\mathcal{G}| \ll T \times |\mathcal{I}|$, and thanks to the sparsity of $A$, the computation of $A\boldsymbol{\mu}$ is highly efficient. This operation serves to project the variable $\boldsymbol{\mu}$ from the group space into the item spaces.*

The detailed proof can be seen in the Appendix A.2.

## 4.3 FairSync Algorithm

The Figure 2 shows the framework of the FairSync. FairSync will retrieve items from the transformed dual space. Next, we will illustrate the inference phase and online learning phase, respectively.

*4.3.1 Inference phase.* we will provide illustrative examples to demonstrate how FairSync works and present a detailed inference workflow of FairSync.

Firstly, Figure 2 (a) provides an illustrative example to demonstrate the functioning of the FairSync algorithm in an intuitive manner. In this example, we simplify the problem to retrieving two items from a corpus containing four items (depicted as circles in the figure), each assigned to different groups, represented by distinct colors in the figure. A user $u_t$ arrives at the recommender system, and this user is represented as the red pentagram. The system's requirement is to ensure that there is at least one exposure of each group. From the original space, the user and items are represented as the embedding $\boldsymbol{e}_u, \boldsymbol{e}_i$ in Section 4.1. In such space, the retrieval system will find the top-2 closest items, i.e. orange items to users. However, in the dual space, FairSync will project the user and item

---

**Algorithm 1:** FairSync Algorithm

**Input:** User arriving order $\{u_t\}_{t=1}^{T}$, item corpus $\mathcal{I}$, candidate size $K$, batch size $B$, optimizer Opt with learning rate $\eta$, trained user item embedding network $m^u(\cdot), m^i(\cdot)$ item-group adjacent matrix $A$, minimum group exposure requirement $\{m_g\}_{g \in \mathcal{G}}$.

**Output:** The candidate lists for every user $\{L_K(u_t)\}_{t=1}^{T}$

1: Calculate items embeddings $\{\boldsymbol{e}_i = m^i(i), \forall i \in \mathcal{I}\}$
2: Re-construct and distributively index the item embeddings $\{\boldsymbol{h}_i = \boldsymbol{e}_i \| A_i, \forall i \in \mathcal{I}\}$ utilizing the Equation (5).
3: Initialize update count $b = 0$.
4: Initialize the gradient buffer $\mathcal{B}_s = \{\}$.
5: **for** $t = 1, \cdots, T$ **do**
6:    Initialize dual solution $\boldsymbol{\mu} = 0$
7:    User $u_t$ arrives
8:    Calculate user embedding $\boldsymbol{e}_{u_t} = m^u(u_t)$
9:    Re-construct query embeddings $\boldsymbol{q}_{u_t} = \boldsymbol{e}_{u_t} \| \boldsymbol{\mu}$ utilizing the Equation (5)
10:    // KNN Retrieval: (Equation (4))
11:    $L_K(u_t) = \arg\min_{S \subset \{1,2,...,|\mathcal{I}|\}, |S|=k} \sum_{i \in S} d^{Dual}(\boldsymbol{q}_{u_t}, \boldsymbol{h}_i)$,
12:    // Compute the sub-gradient:
13:    Compute the sub-gradient $s$ utilizing the Equation (6)
14:    Store the sub-gradient $s$ into $\mathcal{B}_s$
15:    Update count $c = c + 1$
16:    **if** $c = B$ // Update per B users **then**
17:       $\boldsymbol{u} = \text{Opt}(\boldsymbol{\mu}, \sum_{s \in \mathcal{B}_s} s)$ (Equation (7))
18:       Initialize update count $b = 0$.
19:       Initialize the gradient buffer $\mathcal{B}_s = \{\}$
20:    **end if**
21: **end for**

---

embeddings to different points, while ensuring the minimum exposure constraint is satisfied. In the dual space, the dense retrieval system can efficiently locate the distributed items that meet the requirements and simultaneously maintain retrieval accuracy.

Formally, from Theorem 1, we can observe that the distance between the user and item in the dual space transforms to:

$$d^{Dual} = d(\boldsymbol{e}_u, \boldsymbol{e}_i) + \boldsymbol{\mu}_g, \quad i \in \mathcal{I}_g. \quad (4)$$

Therefore, to better adapt to the dense retrieval architecture discussed in Section 4.1, we reconstruct user $u_t$ embeddings to $\boldsymbol{q}_{u_t}$ and embedding of item $i$ to $\boldsymbol{h}_i$, where $\boldsymbol{q}_{u_t}$ and $\boldsymbol{h}_i$ are defined as follows:

$$\boldsymbol{q}_{u_t} = \boldsymbol{e}_{u_t} \| - \boldsymbol{\mu}_t, \quad \boldsymbol{h}_i = \boldsymbol{e}_i \| A_i, \quad (5)$$

where $\|$ denotes the concat operator between two vectors and $A_i$ denotes the $i$-th column vector of adjacent matrix in Theorem 1. Therefore, we have

$$d^{Dual} = -\boldsymbol{q}_{u_t}^{\top} \boldsymbol{h}_i.$$

Figure 2 (b) illustrate the inference phase of FairSync in a more visualized way. Firstly, a user $u_t$ arrives, then the user embedding extraction $m^u$ module (any retrieval model) will extract the user embedding $\boldsymbol{e}_{u_t}$. Then, at time $t$, we have a dual vector $\boldsymbol{\mu}_t$ to form the query vector $\boldsymbol{q}_{u_t}$ (Equation (5)). Then we will utilized the vector $\boldsymbol{q}_{u_t}$ to utilize k-nearest neighbors (KNN) search on the distributively indexed item embeddings $\{\boldsymbol{h}_i, \forall i \in \mathcal{I}\}$ in the dual space to retrieve

**Table 1: Statistics of the datasets.**

| Dataset | #User | #Item | #Group | #Interaction |
|---|---|---|---|---|
| Amazon-Book | 459,133 | 313,966 | 165 | 8,898,041 |
| Taobao | 976,779 | 1,708,530 | 1246 | 85,384,110 |

a corresponding list of candidate items (Figure 2 (c)).

*4.3.2 Online learning phase.* In the online learning phase, we aim to update the dual vector $\mu_t$ once in a while.

Specifically, we can see that the sub-gradient $s \in \mathbb{R}^{|\mathcal{G}|}$, $s \in \partial W^{\text{Dual}}/\partial \mu_t$ in Equation (3) at time $t$ satisfies:

$$s_g = \begin{cases} m_g + \sum_{i \in L_K(u_t)} I(i \in \mathcal{I}_g), & \text{if } g \neq \hat{g} \\ \sum_{i \in L_K(u_t)} I(i \in \mathcal{I}_g) + (TK - \sum_{g \neq \hat{g}} m_g), & \text{else}, \end{cases} \quad (6)$$

where $I(\cdot)$ denotes the indicator function and

$$\hat{g} = \arg\max_{g \in \mathcal{G}} \mu_g.$$

Based on the assumption that user comes to the system randomly [40], we can utilize the sub-gradient $s$ to update $\mu_t$. In real applications, however, updating $\mu_t$ at every time step $t$ is challenged by a large number of asynchronous update operations on different servers, and when the update frequency is too high, it can lead to excessively long recall times, thereby impacting the user experience. Therefore, to trade-off the efficiency and effectiveness, we will update the dual vector $\mu$ each $B$ steps.

Specifically, we will store the sub-gradient $s$ of each step into a gradient buffer $\mathcal{B}_s$. For each $B$ steps, we will utilized any optimizer Opt (in this paper, we utilized the well-performing Adam [18]) to update $\mu$ utilizing the averaged gradient in the buffer, i.e.

$$\mu = \text{Opt}(\mu, \sum_{s \in \mathcal{B}_s} s). \quad (7)$$

The detailed FairSync algorithm is shown in Algorithm 1.

## 5 EXPERIMENT

We conducted experiments to demonstrate the effectiveness of the proposed FairSync. The source code and experiments have been shared at github[2].

### 5.1 Experimental settings

*5.1.1 Datasets.* Following the practice in Cen et al. [6], the experiments were conducted on one common used publicly available retrieval datasets and one billion-scale industrial dataset, including:

**Amazon-Book**[3]: The subsets (book domains) of Amazon [15] Product dataset. The item grouping relies on the field "categories". Each training sample is truncated at length 20. As a pre-processing step, we consider groups with fewer than 50 items as a single group, which we name the "infrequent group".

**Taobao**[4]: collected about 1 million user behaviors data based on Taobao's recommender systems [46] during November 25 to December 03, 2017. The item grouping relies on the field "category ID". Each training sample is truncated at length 50. As a pre-processing

[2]https://anonymous.4open.science/r/WWW2023-FairSync-72B1
[3]http://jmcauley.ucsd.edu/data/amazon/
[4]https://tianchi.aliyun.com/dataset/dataDetail?dataId=649&userId=1

step, we consider groups with fewer than 200 items as a single group, which we name the "infrequent group".

The statistics of the two datasets are shown in Table 1.

*5.1.2 Evaluation.* Firstly, we follow the common practice of the sequential recommendation retrieval model [6, 20, 21] to get the embedding extraction network $m^i(\cdot), m^u(\cdot)$. We sorted all the interactions in the dataset based on their timestamps and utilized the initial 80% of the interactions as the training data for network $m^i(\cdot), m^u(\cdot)$ training. The remaining 20% of interactions were split into two equal parts, with each 10% portion serving as the validation and test data, respectively, for evaluation.

As for the evaluation metrics, the performances of the models were evaluated from two aspects: retrieval accuracy, the minimum group exposure satisfaction (i.e. performance of fairness). Let $T$ be the test set length and $\hat{\mathcal{I}}_u$ be the set of items for user $u$ in the test set.

As for the retrieval accuracy, following the practices in [6], we utilized

- **Recall**. We adopt per-user average instead of global average [6, 7]:

$$\text{Recall@N} = \frac{1}{T} \sum_{t=1}^{T} \frac{|L_K(u_t) \cap \hat{\mathcal{I}}_{u_t}|}{\hat{\mathcal{I}}_{u_t}}.$$

- **Hit Rate.** The HR Rate (HR) is a metric that quantifies the percentage of recommended items that include at least one item that the user has previously interacted with [6, 7].

$$\text{HR@N} = \frac{1}{T} \sum_{t=1}^{T} I(|L_K(u_t) \cap \hat{\mathcal{I}}_{u_t}| > 0).$$

- **Normalized Discounted Cumulative Gain.** Normalized Discounted Cumulative Gain (NDCG) is a metric that factors in the positions of correctly recommended items, providing a measure that accounts for the item's relevance and its position in the recommendation list [6].

$$\text{NDCG@N} = \frac{1}{T} \sum_{t=1}^{T} \sum_{i \in L_K(u_t)} \frac{I(i \in \hat{\mathcal{I}}_{u_t})}{\log_2(\text{pos}(i, L_K(u_t)))}/Z_t,$$

where $\text{pos}(i, L_K(u_t))$ is the sorting position of item $i$ in the list $L_K(u_t)$, starting from 1 to $K$ and $Z_t$ represents a normalization constant that denotes the ideal discounted cumulative gain (IDCG@N), which signifies the highest achievable value for the numerator in the metric at time $t$.

For the minimum group exposure satisfaction, we apply:

- **Enough Satisfaction Groups.** Enough satisfaction groups (ESP) aims to estimate whether each candidate generation policy selects enough items that satisfies the minimum group exposure requirement, similar to the enough relevant items (ER) metric in [38]:

$$\text{ESP} = \frac{1}{|\mathcal{G}|} \sum_{g \in \mathcal{G}} I\left(\left[\sum_{t=1}^{T} \sum_{i \in L_K(u_t)} I(i \in \mathcal{I}_g)\right] > m_g\right)$$

.

**Table 2: Performance comparisons between ours and the baselines on Amazon book subset and Taobao. Our objective is to guarantee that each group possesses a minimum of 200 exposures to fulfill the ESP metric. The ∗ means the improvements over the baseline that can guarantee minimum exposure baselines (K-neighbor and Uncalibrated) are statistical significant (t-tests and $p$-value $< 0.05$). The bold number indicates that the accuracy value exceeds that of all the baselines. All the numbers in the table are percentage numbers with "%" omitted.**

| Base model | Fairness model | Amazon-Book dataset | | | | | | | | Taobao dataset | | | | | | | |
|---|---|---|---|---|---|---|---|---|---|---|---|---|---|---|---|---|---|
| | | top-20 | | | | top-50 | | | | top-20 | | | | top-50 | | | |
| | | Recall | NDCG | HR | ESP | Recall | NDCG | HR | ESP | Recall | NDCG | HR | ESP | Recall | NDCG | HR | ESP |
| youtubeDNN | regularized-fair | 4.52 | 4.61 | 10.13 | 53.94 | 7.11 | 5.64 | 15.55 | 81.21 | 3.29 | 14.85 | 28.89 | 58.27 | 4.97 | 16.56 | 39.31 | 82.83 |
| | IPW | 4.55 | 4.64 | 10.19 | 45.45 | 7.16 | 5.68 | 15.66 | 73.94 | 3.29 | 14.85 | 28.89 | 57.78 | 4.97 | 16.56 | 39.31 | 82.66 |
| | K-neighbor | 0.09 | 0.14 | 0.29 | 100.00 | 0.14 | 0.17 | 0.41 | 100.00 | 0.15 | 0.87 | 1.73 | 100.00 | 0.24 | 1.00 | 2.51 | 100.00 |
| | Uncalibrated | 4.44 | 4.53 | 9.96 | 100.00 | 7.08 | 5.62 | 15.51 | 100.00 | 2.99 | 13.46 | 26.18 | 100.00 | 4.79 | 15.95 | 37.87 | 100.00 |
| | **FairSync(ours)** | 4.55* | 4.64* | 10.19* | 100.00 | 7.16* | 5.69* | 15.68* | 100.00 | 3.29* | 14.77* | 28.74* | 100.00 | 4.99* | 16.56* | 39.32* | 100.00 |
| GRU4REC | regularized-fair | 3.95 | 4.01 | 8.70 | 46.67 | 6.35 | 4.94 | 13.63 | 77.58 | 4.73 | 18.84 | 35.63 | 64.93 | 6.95 | 20.43 | 45.99 | 83.63 |
| | IPW | 3.97 | 4.04 | 8.76 | 38.79 | 6.38 | 4.97 | 13.70 | 63.03 | 4.73 | 18.84 | 35.64 | 64.69 | 6.95 | 20.43 | 45.99 | 83.55 |
| | K-neighbor | 0.09 | 0.13 | 0.26 | 100.00 | 0.14 | 0.15 | 0.41 | 100.00 | 0.17 | 0.79 | 1.54 | 100.00 | 0.24 | 0.92 | 2.19 | 100.00 |
| | Uncalibrated | 3.90 | 3.94 | 8.58 | 100.00 | 6.32 | 4.91 | 13.55 | 100.00 | 4.29 | 17.08 | 32.26 | 100.00 | 6.69 | 19.65 | 44.25 | 100.00 |
| | **FairSync(ours)** | 3.98* | 4.04* | 8.77* | 100.00 | 6.37* | 4.97* | 13.68* | 100.00 | 4.74* | 18.79* | 35.52* | 100.00 | 6.96* | 20.54* | 46.01* | 100.00 |
| MIND | regularized-fair | 6.64 | 6.58 | 13.70 | 41.82 | 9.64 | 7.66 | 19.46 | 63.64 | 4.62 | 18.98 | 36.15 | 62.28 | 6.96 | 20.70 | 47.34 | 79.21 |
| | IPW | 6.62 | 6.56 | 13.67 | 38.18 | 9.63 | 7.63 | 19.42 | 58.79 | 4.62 | 18.98 | 36.15 | 62.28 | 6.96 | 20.70 | 47.34 | 78.97 |
| | K-neighbor | 0.10 | 0.16 | 0.32 | 100.00 | 0.15 | 0.18 | 0.40 | 100.00 | 0.17 | 0.90 | 1.80 | 100.00 | 0.26 | 1.12 | 2.60 | 100.00 |
| | Uncalibrated | 6.45 | 6.39 | 13.33 | 100.00 | 9.52 | 7.54 | 19.20 | 100.00 | 4.20 | 17.23 | 32.80 | 100.00 | 6.69 | 19.93 | 45.58 | 100.00 |
| | **FairSync(ours)** | 6.60* | 6.60* | 13.65* | 100.00 | 9.65* | 7.69* | 19.48* | 100.00 | 4.57* | 18.82* | 35.86* | 100.00 | 6.98* | 20.76* | 47.38* | 100.00 |
| ComiRec-DR | regularized-fair | 4.92 | 5.26 | 10.99 | 37.58 | 7.40 | 6.20 | 16.03 | 61.21 | 5.51 | 23.49 | 42.25 | 63.24 | 7.98 | 24.85 | 52.77 | 80.26 |
| | IPW | 4.91 | 5.24 | 10.97 | 33.33 | 7.41 | 6.18 | 16.03 | 55.15 | 5.51 | 23.49 | 42.25 | 63.24 | 7.98 | 24.85 | 52.76 | 80.26 |
| | K-neighbor | 0.09 | 0.14 | 0.25 | 100.00 | 0.14 | 0.16 | 0.37 | 100.00 | 0.19 | 1.01 | 1.85 | 100.00 | 0.28 | 1.18 | 2.60 | 100.00 |
| | Uncalibrated | 4.76 | 5.10 | 10.68 | 100.00 | 7.30 | 6.10 | 15.82 | 100.00 | 4.99 | 21.29 | 38.30 | 100.00 | 7.67 | 23.92 | 50.81 | 100.00 |
| | **FairSync(ours)** | 4.92* | 5.28* | 11.0* | 100.00 | 7.42* | 6.20* | 16.08* | 100.00 | 5.47* | 23.35* | 42.20* | 100.00 | 8.07* | 24.93* | 52.80* | 100.00 |
| ComiRec-SA | regularized-fair | 5.23 | 3.78 | 10.83 | 49.70 | 8.09 | 4.93 | 16.47 | 75.76 | 5.49 | 23.77 | 41.61 | 63.88 | 7.76 | 24.98 | 51.28 | 80.10 |
| | IPW | 5.25 | 3.79 | 10.85 | 44.85 | 8.10 | 4.93 | 16.46 | 70.91 | 5.49 | 23.77 | 41.62 | 63.80 | 7.76 | 24.99 | 51.28 | 80.10 |
| | K-neighbor | 0.11 | 0.14 | 0.29 | 100.00 | 0.15 | 0.75 | 1.92 | 100.00 | 0.17 | 0.90 | 1.61 | 100.00 | 0.25 | 1.10 | 2.39 | 100.00 |
| | Uncalibrated | 5.12 | 3.70 | 10.59 | 100.00 | 8.01 | 4.88 | 16.30 | 100.00 | 4.97 | 21.53 | 37.65 | 100.00 | 7.47 | 24.06 | 49.36 | 100.00 |
| | **FairSync(ours)** | 5.26* | 3.80* | 10.81* | 100.00 | 8.12* | 4.93* | 16.47* | 100.00 | 5.45* | 23.66* | 41.36* | 100.00 | 7.76* | 24.99* | 51.33* | 100.00 |

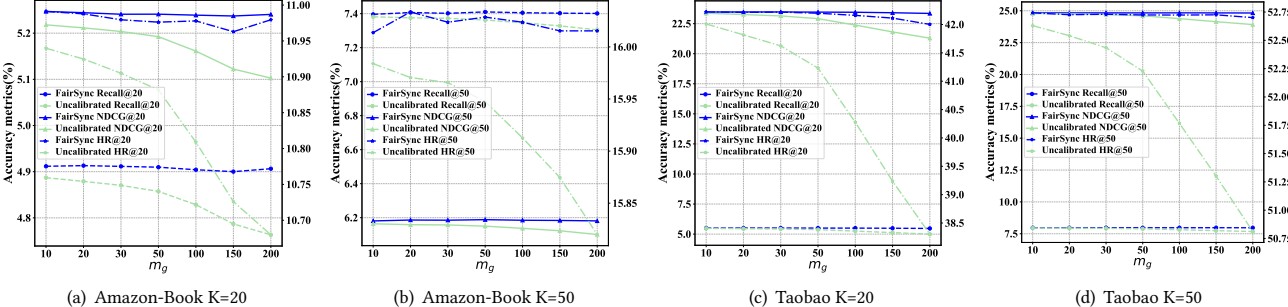

| (a) Amazon-Book K=20 | (b) Amazon-Book K=50 | (c) Taobao K=20 | (d) Taobao K=50 |
|---|---|---|---|

**Figure 3: The accuracy curve (Recall, NDCG, and HR) of FairSync (ours) and the best baseline Uncalibrated under different minimum exposure threshold $m_g$. The experiments were conducted based on the best retrieval base model ComiRec-DR.**

*5.1.3 Baselines and Base Models.* In this section, we mainly introduce the controllable retrieval baselines and base models used for extraction user and item embeddings.

For the distributed retrieval baseline, we mainly choose four heuristic methods: **regularized-fair** (detailed pseudo-code in Appendix B): at each time step $t$, a regularized-based dual variable $\mu_r$ is replaced with the dual variable $\mu$ of FairSync, which aims to reduce the exposure gaps between the all items and the worst-off item. **IPW** [38]: selected the group exposure as item's inverse propensity weighted (IPW) during the retrieval process. However, the aforementioned two baselines are not able to ensure the required minimum exposures of groups.

The next baselines are the two heuristic methods used to ensure that the required minimum exposures of groups are guaranteed in the retrieval process. $K$-**neighbor** [31]: at each time step $t$, Only the items on each server associated with the top-K group, having the lowest cumulative exposure, are retrieved. **Uncalibrated** [38]: each step $t$ only choose the items whose group do not satisfy the required exposures. For items that already meet the required exposure criteria, we retrieve them using the KNN search method as the base models.

For the retrieval base models, we utilize: **Youtube DNN** [9]: the most commonly used retrieval models in industrial recommender systems; **GRU4Rec** [16]: utilized the recurrent neural network (RNN) to model the user sequential behaviors in the retrieval process; **MIND** [21]: aimed to model user's diverse interests by designing a multi-interest extractor layer based on the capsule

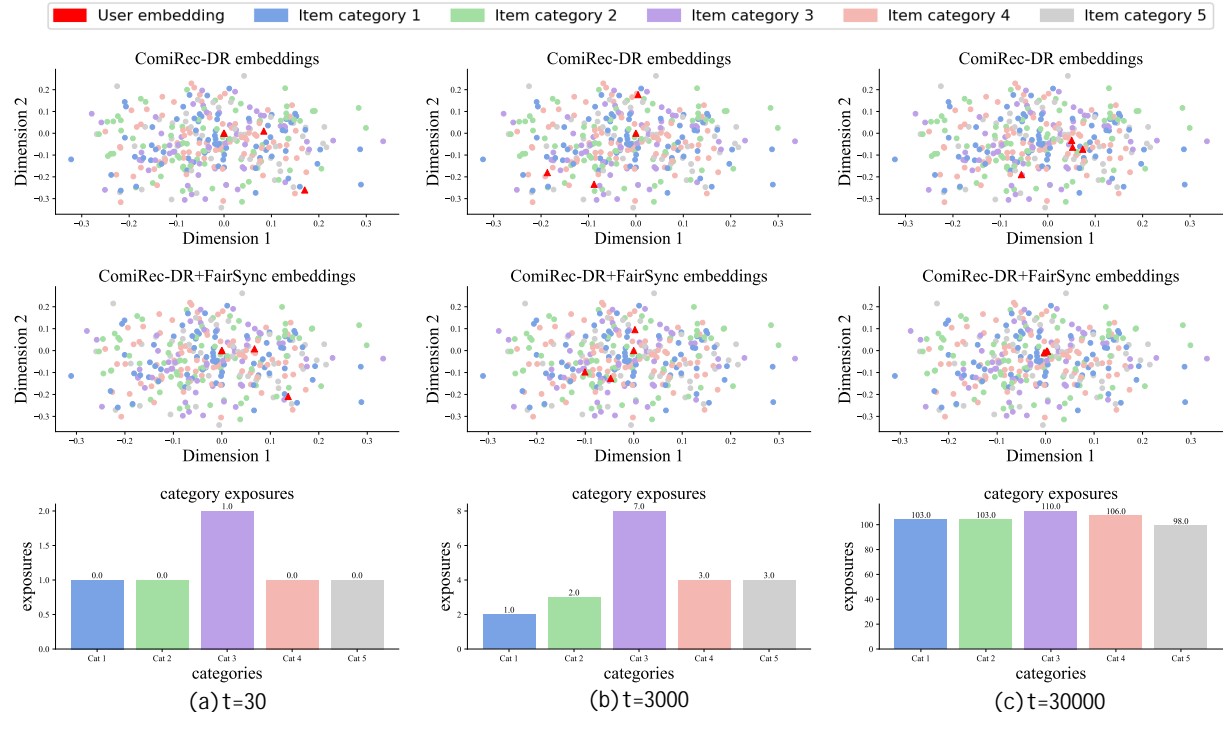

Figure 4: The three sub-figures in the first row illustrate the t-SNE visualization of Comirec-DR item embeddings $e_i$ and user embeddings $e_{u_t}$ under different $t$. The three sub-figures in the second row depict the t-SNE visualization of our model FairSync using Comirec-DR as the base model for item embeddings $h_i$ and user embeddings $q_{u_t}$ under different $t$. The three sub-figures in the final row depict the category exposures under different time steps $t$. The experiment was conducted on the Amazon-book dataset with retrieval number $K = 50$.

routing mechanism [13]; **ComiRec-SA** [6]: the recent state-of-the-art retrieval models, which captured user diverse interests by the self-attention mechanism. **ComiRec-DR** [6]: the variant of ComiRec-SA, which used the dynamic routing method to model user's sequential behaviors.

*5.1.4 Implementation details.* As for the hyper-parameters in all models, the learning rate $\eta$ was tuned among $[1e - 2, 1e - 4]$ and the batch size for updating dual vector $B$ was tuned among $[1, 512]$. For training the base retrieval model, we utilized the best parameters reported in the original papers of the models. We implemente FairSync with the most common faiss [17] KNN-search package. The gradient descent package used Pytorch [30] to apply the auto-gradient. The experiments were conducted under a server with a single NVIDIA GeForce RTX 3090.

## 5.2 Experimental Results on Full Datasets

In this section, we conduct experiments using our model FairSync along with other baselines across all datasets to validate the effectiveness of FairSync.

Firstly, we conduct experiments to show the performance of FairSync and other baselines under the same minimum exposure requirement ($m_g = 200, \forall g \in \mathcal{G}$) across all retrieval base models. Table 2 presents the experimental outcomes for our FairSync model and the baseline methods across all datasets, while ensuring that

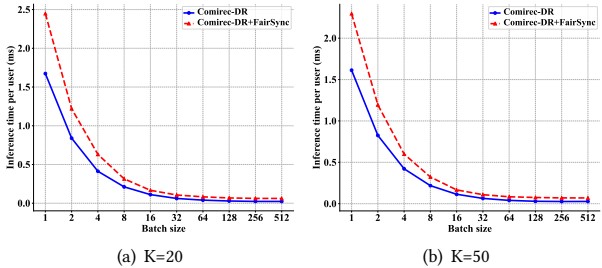

Figure 5: Inference time per user w.r.t. w.r.t. batch size $B$.

each groups maintains a minimum of 200 exposures as a requirement. To make fair comparisons, all the baselines were tuned their hyper parameters to obtain the best performance under our settings.

Based on the reported findings, it becomes evident that our model FairSync effectively fulfills the requirement of each groups maintaining a minimum of 200 exposures (i.e., ESP=100%). Furthermore, FairSync significantly outperform the baseline techniques intended for guaranteeing minimum exposure (K-neighbor and Uncalibrated) across all datasets and various base retrieval models, encompassing different top-K retrieval numbers, as reflected in accuracy metrics including Recall, NDCG, and HR. Simultaneously, FairSync exhibits accuracy performance that is comparable with

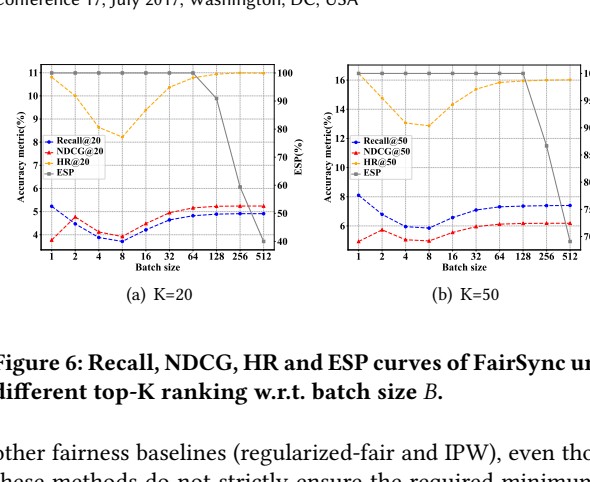

(a) K=20

(b) K=50

**Figure 6: Recall, NDCG, HR and ESP curves of FairSync under different top-K ranking w.r.t. batch size $B$.**

other fairness baselines (regularized-fair and IPW), even though these methods do not strictly ensure the required minimum exposure of groups. The experiments conclusively demonstrate that FairSync effectively guarantees the minimum exposure requirement without significantly compromising accuracy of retrieval process.

Secondly, we conduct experiments to demonstrate the performance of FairSync and the best baseline (Uncalibrated) under varying minimum exposure requirements under the best retrieval base model ComiRec-DR. Figure 3 reports the accuracy (Recall, NDCG and HR) curve of our model FairSync and the best baseline Uncalibrated under different minimum exposure threshold $m_g \in [10, 200]$, $\forall g \in \mathcal{G}$. Both FairSync and Uncalibrated are able to satisfy the minimum exposure requirements.

From the curves presented in Figure 3 (a-d), it is evident that our model FairSync consistently outperforms Uncalibrated with a large marign across various accuracy metrics, datasets, and retrieval numbers ($K = 20, 50$). The experiment demonstrates that our model FairSync consistently exhibits better accuracy when ensuring the minimum exposure requirements of different groups.

## 5.3 Experiment analysis

We also conduct experiments to analyze FairSync on Amazon-book dataset under the best retrieval base model ComiRec-DR.

*5.3.1 Visualization of embeddings under original and dual space.* In this section, we aim to visualize and illustrate the effective dual projection of FairSync (with ComiRec-DR as base model, i.e. ComiRec+FairSync) by randomly sampling 5 item categories and set the requirement $m_g = 200$, $\forall g \in \mathcal{G}$. Figure 4 utilized t-SNE [36] to visually represent user and item embeddings $\boldsymbol{e}_{u_t}$ and $\boldsymbol{e}_i$ in the original space (sub-figures in first rows), as well as user and item embeddings $\boldsymbol{q}_{u_t}$ and $\boldsymbol{h}_i$ in the dual space (sub-figures in second rows), across various time steps $t$. We also show the category exposures under different time steps $t$ (sub-figures in third rows). Note that ComiRec-DR is a multi-interest retrieval model [6, 21], where we set four user embedding generated to represent different user interests per time step $t$.

Figure 4 (a) illustrates that at the initial retrieval process ($t = 30$), the exposure levels for various categories (as depicted in the third column's bar plots) are nearly equalized. Such equalized exposure, in turn, leads to FairSync's reconstructed embeddings in the dual space (ComiRec-DR+FairSync embeddings) closely mirroring the patterns of the original embeddings (ComiRec-DR embeddings) to maintain retrieval accuracy.

Figure 4 (b, c) illustrates the intermediary and ending stage ($t = 3000, 30000$) of stage-1, during which the category 3 and 4 experiences dominance in exposure levels, whereas the other categories exhibits a lower level of exposure. In the original space (ComiRec-DR embeddings), it is evident that the user embeddings are closely aligned with the embeddings of category 3 and 4, resulting in the dominance of category 3 and 4. However, in the dual space (ComiRec-DR+FairSync embeddings), the user embeddings are in closer proximity to other categories (1,2 and 5), thereby ensuring that other categories meets the minimum exposure requirements. At the same time, the dual embeddings maintain close proximity to the original embeddings, ensuring the preservation of retrieval accuracy.

The experiment clearly demonstrated that throughout the retrieval process, our model FairSync dynamically adjusts the user embedding's position based on category exposure, enhancing retrieval accuracy while maintaining the minimum exposure requirement.

*5.3.2 Ablation study on batch size.* In this section, we aim to conduct experiments to show the performance and inference time influenced by different online batch size $B$, since $B$ controls the dual vector $\boldsymbol{\mu}$'s updating frequency. Figure 5 and Figure 6 depict the variations in inference time and performance, respectively, with respect to the batch size $B \in [1, 512]$.

Firstly, Figure 5 illustrates that the online inference time per user w.r.t. batch size under different retrieval number $K$. From the displayed curve, it is evident that when the batch size is smaller ($B \le 8$), FairSync still demands approximately $[0.2, 1]$ ms more time in comparison to the base model. When the batch size is relatively large ($B > 8$), the inference times of both FairSync and the base model are comparable, typically remaining below 0.25 ms. This satisfies the inference time requirements for industrial applications.

Secondly, Figure 6 illustrates that accuracy (Recall, NDCG and HR) curve and ESP ($m_g = 200$) curve w.r.t. batch size under different retrieval number $K$. Based on the depicted curve, it is apparent that the retrieval accuracy curve decreases as the batch size varies within the range $B \in [1, 8]$, whereas for batch sizes within the range $B \in [8, 512]$, the accuracy curve exhibits an increase. It is also worth noting that the minimum exposure requirement is no longer satisfied as the batch size increases beyond $B > 64$.

Therefore, we observe that the online batch size $B$ is a trade-off co-efficient for performance and inference time. In real-world applications, we must carefully control the online batch size $B$, as larger values can reduce inference time but may result in poorer performance, while smaller values can have the opposite effect.

## 6 CONCLUSION

This paper proposes a novel retrieval model called FairSync that aims to maintain accuracy while ensuring the minimum exposure for specific groups in distributed retreival process. In FairSync, we transform the problem into a constrained distributed optimization problem. and resolved the issue in the dual space of the problem in a distributed manner. Extensive experiments conducted on two large-scale datasets consistently showcased FairSync's superior performance over baseline models across various retrieval base models. Importantly, FairSync manages to maintain minimal computational costs in real-world applications.

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

**Algorithm 2:** regularized-fair Algorithm

---

**Input:** User arriving order $\{u_t\}_{t=1}^T$, item corpus $\mathcal{I}$, candidate size $K$, batch size $B$, trained user item embedding network $m^u(\cdot), m^i(\cdot)$ item-group adjacent matrix $\mathbf{A}$, trade-off coefficient $\lambda$.

**Output:** The candidate lists for every user $\{L_K(u_t)\}_{t=1}^T$

1: Calculate items embeddings $\{e_i = m^i(i), \forall i \in \mathcal{I}\}$
2: Re-construct and distributively index the item embeddings $\{h_i = e_i \| A_i, \forall i \in \mathcal{I}\}$ utilizing the Equation (5).
3: **for** $t = 1, \cdots, T$ **do**
4:    User $u_t$ arrives
5:    Calculate user embedding $e_{u_t} = m^u(u_t)$
6:    Compute $\mu_r = \lambda[\mathbf{A}(e - (\min_{g \in \mathcal{G}} e_g \mathbf{1}^\top)]$
7:    Re-construct query embeddings $q_{u_t} = e_{u_t} \| - \mu_r$.
8:    // KNN Retrieval: (Equation (4))
9:    $L_K(u_t) = \arg\min_{S \subset \{1,2,...,|\mathcal{I}|\}, |S|=k} \sum_{i \in S} d^{\text{Dual}}(q_{u_t}, h_i)$,
10: **end for**

# A APPENDIX

## A.1 Lemma 1

Firstly, we prove a lemma before we start the proof of Theorem 1.

LEMMA 1. *Let $a_{[i]}$ denotes the $i$-th largest element of $a$. Considering the function with the $x \in \mathbb{R}^N$ as the input,*

$$Top\text{-}K(x) = \sum_{k=1}^K x_{[k]}.$$

*We will demonstrate that the function $f(x)$ exhibits concavity with respect to $x$.*

PROOF. By the definition, for any $0 \le \lambda \le 1$ we have

$$\text{Top-K}(\lambda x + (1-\lambda)y) = \sum_{k=1}^K (\lambda x + (1-\lambda)y)_{[k]}$$

$$\le \lambda \sum_{k=1}^K x_{[k]} + (1-\lambda) \sum_{k=1}^K y_{[k]}$$

$$= \lambda \text{Top-K}(x) + (1-\lambda)\text{Top-K}(y).$$

, that is the sum of the first k elements of two vectors added together is less than the sum of the first k elements of the two vectors individually added.

Q.E.D. □

## A.2 Proof of Theorem 1

PROOF. We will firstly list some notations used in our proof: $a_{[i]}$ denotes the $i$-th largest element of $a$. $A_i$ denote the $i$-th column of the matrix $A$.

We can utilize the Lagrangian condition to re-write the minimum guarantee condition $e_g = \sum_{t=1}^T \sum_{i \in \mathcal{I}_g} x_{u_t,i}$ of problem (2) as the following problem:

$$W^{Dual} = \max_{x_{u_t,i}} \min_{\mu} \sum_{t=1}^T \sum_{i=1}^{|\mathcal{I}|} \left[ (r_{u_t,i} - A_i^\top \mu) x_{u_t,i} + \sum_{g \in \mathcal{G}} \mu_g e_g \right]$$

$$\text{s.t.} \quad \sum_{i \in \mathcal{I}} x_{u_t,i} = K, \quad \forall t \in [1,2,\ldots,T]$$

$$x_{u_t,i} = \{0,1\}, \quad \forall i \in \mathcal{I}, \forall t \in [1,2,\ldots,T]$$

$$e_g \ge m_g, \quad \forall g \in \mathcal{G}$$

. Where the $\mu \in \mathbb{R}^{|\mathcal{G}|}$ is the dual vector.

Let's consider the following program:

$$\max_x \quad a^\top x$$

$$\text{s.t.} \quad \mathbf{1}^T x = K, \qquad (8)$$

$$0 \le x \le 1,$$

This problem is a well-studied knapsack problem [33], whose optimal objective should be $\sum_{i=1}^K a_{[i]}$. The equation tells us that only the top $K$ items that user $u_t$ have the highest preference for the every group $p$ will be recommended for every user.

Thus, we can easily observe that the objective $W$ of the target about $x_{t,i}$ is a top-K function in lemma 1 and from lemma 1, we can observe that $W$ is is concave with respect to $x$ and convex with respect to the variable $\mu$. From the minimax theorem [11], we can re-write the equation as:

$$W = \min_\mu \max_{e_g} \quad \sum_{t=1}^T \sum_{k=1}^K \left[ (r_{u_t,i} - A_i^\top \mu)_{[k]} + \sum_{g \in \mathcal{G}} \mu_g e_g \right]$$

$$\text{s.t.} \quad e_g \ge m_g, \quad \forall g \in \mathcal{G} \qquad , \quad (9)$$

$$\sum_{g \in \mathcal{G}} e_g = TK$$

where we replace the variable $x$ to $e$ utilizing the condition $e_g = \sum_{t=1}^T \sum_{i \in \mathcal{I}_g} x_{u_t,i}$. Now, considering the following problem:

$$L = \max_e \quad \sum_{g \in \mathcal{G}} \mu_g e_g$$

$$\text{s.t.} \quad \sum_{g \in \mathcal{G}} e_g = TK,$$

$$e_g \ge m_g, \quad \forall g \in \mathcal{G}$$

, which is a well-studied knapsack problem [33], with the optimal solution

$$\sum_{g \in \mathcal{G}} m_g \mu_g + \max_{g \in \mathcal{G}} \{\mu_g\}(TK - \sum_{g \in \mathcal{G}} m_g).$$

Finally, we can take the optimal solution into Equation (9), we get $W^{Dual}$ as

$$min_\mu \left[ \sum_{t=1}^T \sum_{k=1}^K (r_{u_t,i} - A\mu)_{[k]} + \sum_{g \in \mathcal{G}} m_g \mu_g + \max_g \{\mu_g\}(TK - \sum_{g \in \mathcal{G}} m_g) \right].$$

Q.E.D.

□

# B REGULARIZED-FAIR ALGORITHM

In this section, we propose a heuristic method for distributed approach for improving the worst-off group exposures in retrieval process, aligning with the concept amortized max-min fairness [10, 40], named regularized-fair. Similar with the dual form of FairSync, it introduced a dual variable $\mu_r$ that measures the exposure gaps between the target group and the worst-groups. The detailed algorithm is shown in Algorithm 2.

