# OpenReview forum: "FairSync: Ensuring Amortized Group Exposure in Distributed Recommendation Retrieval"
_ACM.org/TheWebConf/2024/Conference — TheWebConf24_

### Official Review · Reviewer_9dFZ · 2023-11-23

**Novelty:** 5
**Technical Quality:** 5

**Review:**

## Summary

The paper proposed a new method to provide better group exposure in the recommendation system with updates in the retrieval stage. The texts are easy to follow and the method proposed seems to be novel and well-descripted. Overall, I believe the paper meets the criteria of the conference with some comments below.


## Strengths

S1 - Points out the current weakness of the existing group exposure methods as they are more focused on the ranking stage.
S2 - Provides a thorough description of the novel method which could ensure the exposure in the retrieval stage.
S3 - The figures in the paper provide a good overview of the data and are mostly easy to follow.

## Weaknesses

W1 - Some figures can be relocated to another place for better readability.
W2 = the experiment process could be further described.

**Questions:**

## Suggestions for improvements

- In section 2, the paper mainly discussed the related work that focuses on stage 2 to improve group exposure. As the topic of the paper is focused on stage 1 update, it would be better if there could be more introduction about the existing methods of stage 1 in the related work section.

- Within section 4.3, the paper referred to Figure 2 multiple times. However, figure 2 is on the previous page and it has not been referred to on that page. Due to that reason, it will be better to relocate Figure 2 to the page of section 4.3, which could largely improve the readability of that part.

- Section 5.1 points out that the experiment uses the datasets of Amazon-Book and Taobao for testing, besides, it also mentions that the pre-defined categories in the datasets also play an important role. Due to that reason, it is a bit confusing for audiences as there seems to be no discussion about whether these two different datasets are equivalent in the experiment.

- The experiment results have been discussed in section 5.2, which also referred to the multiple results of the baseline results. Currently, there seems not to be so much description about the comparison of the results between the results of the baseline and the new method.

# Questions

- What caused the differences in the performances of the models on the Amazon-Book and Taobal datasets?

**Reviewer Confidence:**

2: The reviewer is willing to defend the evaluation, but it is likely that the reviewer did not understand parts of the paper

**Scope:**

3: The work is somewhat relevant to the Web and to the track, and is of narrow interest to a sub-community

---

### Official Review · Reviewer_Mn9G · 2023-11-24

**Novelty:** 6
**Technical Quality:** 5

**Review:**

The article focuses on stage 1 of the two stage process of displaying recommendations to a user: the retrieval of candidate items from a distributed system of servers, so that in stage-2 these items can be ranked for presentation. The authors argue that most fair ranking work has focused on stage 2 even though the pool of candidates for sorting is heavily influenced by the retrieval in stage 1. To address this gap, the authors introduce a system for ensuring equitable group exposure at the retrieval level. The proposed system is tested on standard benchmarks datasets and it outperforms other fairness approaches at a small cost of computation time.

The article is difficult to parse at times because of small editing problems (or details that are not clarified enough for me to understand):
- in Remark 1 (Distributed solution in dual space). After the transformation of the original problem (Equation (1)) into its dualform (Equation (1)) -> the second equation should be Equation 2?
- in the description of Figure 4 authors write "Figure 4(a) illustrates that at the initial retrieval process (t = 30), the exposure levels for various categories (as depicted in the third column’s bar plots) are nearly equalized." Figure 4a is for t=30 but is not the third column and the exposure levels are not equalized? Cat 3 has exposure 1.0 (but the x-axis says 2.0?) and others have exposure 0. Then the authors write that in 4b,c Cat 3 and Cat 4 experience dominance in exposure levels but that does not track with the histograms in b) (it does track with c)). Then "In the original space (ComiRec-DR embeddings), it is evident that the user embeddings are closely aligned with the embeddings of category 3 and 4, resulting in the dominance of category 3 and 4." How is it evident in the figure? The item categories do not seem to be segregated on the two dimensions, so I'm not sure how one would judge that. Also Cat 4 is at no point dominant according to the histogram, so I'm not sure how to read this.

**Questions:**

The authors should clarify the description of Figure 4 or perhaps rethink how they are presenting these results. The item embeddings do not change at the three time points but this information is repeated six times but it's not clear why it's included in the first place.

**EDIT**: I read the rebuttal and I'm satisfied with the proposed answers to the issues I raised.

**Ethics Review Description:**

No issues

**Reviewer Confidence:**

1: The reviewer's evaluation is an educated guess

**Scope:**

4: The work is relevant to the Web and to the track, and is of broad interest to the community

---

### Official Review · Reviewer_bGxK · 2023-11-24

**Novelty:** 4
**Technical Quality:** 3

**Review:**

### Summary
The paper tackles the issue of fair recommendation retrieval. While most works focus on the second stage of retrieval in which a short list of candidates have already been selected, this paper focuses on generating a fair shortlist. For this reason, scalability is of particular concern as the set of candidate items is much larger in the first stage. The authors present a scalable algorithm that augments user embeddings with a "dual vector", which essentially provides a handicap for certain item categories. The dual vector can be calculated by optimizing an unconstrained objective. The authors present results that their method increases group exposure at the item level while preserving recommendation accuracy.

### Pros
* The problem definition is well-defined and the goals are made clear.
* The authors discuss scalability in a manner that considers the distributed nature of the servers.
* The algorithm presented is easily integrable on top of existing embedding algorithms and the dual vector is interpretable.

### Cons
* My main concern is that several steps in the proof for Theorem 1 appear to lack justification or make incorrect assumptions. I will elaborate in the questions section. As Theorem 1 introduces the dual objective and is the crux of the paper my assessment will depend heavily on clarifications of the proof.
* The problem setup in section 4.2, specifically Equation 2, considers the offline setting in which the item decisions for all time steps are optimized in one shot. In contrast, FairSync considers a setting in which decisions are made in an online setting one user at a time. It would be helpful to explain the transition from the offline setting to the online setting.
* I appreciate the goal of Figure 4 however the visual illustration of FairSync is not very clear as the top and middle rows are not very distinguishable.

### Miscellaneous
* On page 4 right below Equation 5 it should be the "i-th row" instead of the "i-th" column.
* On page 3 after Equation 1, I would consider d to be a distance measure instead of a distance metric as not all axioms of metrics are satisfied by the dot product.

**Questions:**

* Regarding Theorem 1, I am unsure of the use of the knapsack problem in this setting. Specifically, the application of Equation 8 appears to disregard the exposure values $e_g$ which are in fact defined in terms of $x_{u_t, i}$ and well as the exposure constraints. Given that the exposure values are dependent on $x$ it is not necessarily the case that the top K items will be recommended. Am I missing anything with the knapsack substitution?

* Expanding on the previous "con", it would be helpful if the authors could explain the connection between the offline problem introduced in Equation 2 and the online problem that FairSync solves. Is the dual vector still valid for an individual user when it is originally optimized over a set of users?

**Reviewer Confidence:**

3: The reviewer is confident but not certain that the evaluation is correct

**Scope:**

3: The work is somewhat relevant to the Web and to the track, and is of narrow interest to a sub-community

---

### Official Review · Reviewer_sFmC · 2023-12-01

**Novelty:** 5
**Technical Quality:** 5

**Review:**

The paper proposes FairSync which is a retrieval method for ensuring that group exposure constraints can be satisfied in an amortized fashion when doing distributed retrieval in recommender systems. The problem is important because currently there isn't a known practice for implementing constrained nearest neighbor retrieval (say for fairness constraints). The method is very intuitive and straightforward though -- the authors translate the constrained nearest neighbor problem to an unconstrained dual form that works well with distributed retrieval. The dual form is also pretty intuitive -- each user embedding is appended with the current exposure info for different groups, and each item embedding (stored on the clusters) is appended with the group information of the items.
The authors show, through extensive experiments and comparisons with baseline algorithms, that Fairsync satisfies exposure constraints while maintaining high recall, NDCG, hit rate metrics.

Some potential weaknesses and limitations of the work are:
- The details for the distributed retrieval is fairly limited. The paper does mention that it is the distributed dense retrieval architecture but that is also pretty generic, and does not say much about other systems that might handle indexing and serving differently.
- What happens when the embedding space has a lot of very homogenous neighborhoods, i.e., the nearest neighborhoods of user embeddings are mostly composed of the same group, and searching for items from underrepresented groups can be very expensive in such an embedding. If the embedding homogeneity indeed affects performance of retrieval, it might be worth studying it to test for extremes on a toy dataset/embedding.
- The paper only considers a single kind of fairness constraint, where the exposure if constrained globally. How does the method work when the retrieved set may need to be controlled at a per query level, i.e., different users have different demand for diversity in the retrieved set.
- The baselines are not described with enough detail. Please provide better descriptions of the baseline methods in the main text or the appendix. For example, without reading the cited paper, it is not clear how the IPW method does not ensure that the constraint is satisfied.
- The paper is technically well written but has a bunch of typos and grammatical errors affecting readability. e.g.
-- Page 7, Section 5.1.3 Baselines and Base Models, first paragraph: "We implemente FairSync" missing "d" at the end of implemente.
-- Page 8, Figure 4 caption: "Figure 4 utilized t-SNE" should be "utilizes" or "uses" instead of "utilized".
-- Page 3, Section 4.1 Extra comma after ei should be removed.
-- Line 355, "can be written as" instead of "can be write as"
- The problem tackled in this paper is motivated in a recent industry paper, and a solution similar to one of the baselines (kNN method?) is also presented in: Representation Online Matters: Practical End-to-End Diversification in Search and Recommender Systems (Silva et al. 2023).

Overall, the paper studies an important and somewhat ignored problem in the space of fairness for recommender systems. The simplicity of the solution lends itself to practical impact in real world recommender systems, and hence the paper could be a good contribution to the community. However, the authors are encouraged to consider the feedback above.

**Questions:**

The questions to the authors are mentioned in the limitations and weaknesses part of the review above. Here are short summaries of the question:
- (How) does the method apply to other distributed retrieval architectures?
- What happens in the extreme case when the constraints might not be satisfiable? or the constraints cost a lot in terms of latency or recall?
- How do query level constraints work with this method?
- Can you clarify the baselines, and how some of them may or may not satisfy the constraint?

**Reviewer Confidence:**

4: The reviewer is certain that the evaluation is correct and very familiar with the relevant literature

**Scope:**

4: The work is relevant to the Web and to the track, and is of broad interest to the community

---

### Decision · Program_Chairs · 2024-01-22

**Decision:**

Accept

**Comment:**

Our decision is to accept. Please see the AC's review below and improve the work considering that and the reviewers' feedback for cemera-ready submission.

"Reviewers generally praised this paper as studying an important problem relevant to TheWebConf, and producing a method that could be applied in real recommender systems. While some concerns were raised about the correctness of the theory in the paper, these seem to have been resolved in the rebuttal period. I think this paper provides incremental value to the conference and recommend it is accepted."